# Vandetanib versus Cabozantinib in Medullary Thyroid Carcinoma: A Focus on Anti-Angiogenic Effects in Zebrafish Model

**DOI:** 10.3390/ijms22063031

**Published:** 2021-03-16

**Authors:** Silvia Carra, Germano Gaudenzi, Alessandra Dicitore, Davide Saronni, Maria Celeste Cantone, Alice Plebani, Anna Ghilardi, Maria Orietta Borghi, Leo J. Hofland, Luca Persani, Giovanni Vitale

**Affiliations:** 1Laboratory of Endocrine and Metabolic Research, Istituto Auxologico Italiano, IRCCS, 20095 Cusano Milanino, MI, Italy; carra-silvia@libero.it (S.C.); luca.persani@unimi.it (L.P.); 2Laboratory of Geriatric and Oncologic Neuroendocrinology Research, Istituto Auxologico Italiano, IRCCS, 20095 Cusano Milanino, MI, Italy; germano.gaudenzi@gmail.com (G.G.); alessandra.dicitore@libero.it (A.D.); plebanialice94@gmail.com (A.P.); 3Department of Medical Biotechnology and Translational Medicine, University of Milan, 20122 Milan, Italy; davide.saronni1@gmail.com (D.S.); celeste.cantone@gmail.com (M.C.C.); 4Department of Biosciences, University of Milan, 20133 Milan, Italy; anna.ghilardi@unimi.it; 5Experimental Laboratory of Immuno-Rheumatology, Istituto Auxologico Italiano, IRCCS, 20095 Cusano Milanino, MI, Italy; maria.borghi@unimi.it; 6Department of Clinical Sciences and Community Health, University of Milan, 20122 Milan, Italy; 7Division of Endocrinology, Department of Internal Medicine, Erasmus MC, 3015 GD Rotterdam, The Netherlands; l.hofland@erasmusmc.nl

**Keywords:** medullary thyroid carcinoma (MTC), zebrafish, tumor xenograft, tyrosine kinase inhibitors (TKIs), angiogenesis, cabozantinib, vandetanib

## Abstract

Medullary thyroid carcinoma (MTC) is a tumor deriving from the thyroid C cells. Vandetanib (VAN) and cabozantinib (CAB) are two tyrosine kinase inhibitors targeting REarranged during Transfection (RET) and other kinase receptors and are approved for the treatment of advanced MTC. We aim to compare the in vitro and in vivo anti-tumor activity of VAN and CAB in MTC. The effects of VAN and CAB on viability, cell cycle, and apoptosis of TT and MZ-CRC-1 cells are evaluated in vitro using an MTT assay, DNA flow cytometry with propidium iodide, and Annexin V-FITC/propidium iodide staining, respectively. In vivo, the anti-angiogenic potential of VAN and CAB is evaluated in *Tg(fli1a:EGFP)^y1^* transgenic fluorescent zebrafish embryos by analyzing the effects on the physiological development of the sub-intestinal vein plexus and the tumor-induced angiogenesis after TT and MZ-CRC-1 xenotransplantation. VAN and CAB exert comparable effects on TT and MZ-CRC-1 viability inhibition and cell cycle perturbation, and stimulated apoptosis with a prominent effect by VAN in MZ-CRC-1 and CAB in TT cells. Regarding zebrafish, both drugs inhibit angiogenesis in a dose-dependent manner, in particular CAB shows a more potent anti-angiogenic activity than VAN. To conclude, although VAN and CAB show comparable antiproliferative effects in MTC, the anti-angiogenic activity of CAB appears to be more relevant.

## 1. Introduction

Medullary thyroid carcinoma (MTC) is a rare neuroendocrine tumor, that arises from calcitonin-producing parafollicular C cells of the thyroid gland [1,2]. Although the majority of MTCs are sporadic, in 25% of patients this malignancy occurs in a hereditary form as the dominant component of the Multiple Endocrine Neoplasia (MEN) type 2 syndromes, MEN2A and MEN2B, or familial MTC [3,4]. Alterations in the *REarranged during Transfection* (*RET*) proto-oncogene represent the most crucial events that lead to the development of MTC. *RET* encodes a transmembrane receptor of the tyrosine kinase family [5,6,7,8,9]. The clinical course of patients with MTC is variable, ranging from mild to extremely aggressive. Occurring at diagnosis, about half of the patients present with an advanced stage (III or IV) [3,10]. Surgery is the only curative treatment for MTC since other therapies, including radiotherapy and chemotherapy, have not demonstrated an improvement in long-term survival [3,11].

Although genetic alterations of *RET* are considered the main event involved in the pathogenesis of the vast majority of MTC cases, other kinase receptors may play an important role in the development and progression of this malignancy. Overexpression of epidermal growth factor receptor (EGFR), vascular endothelial growth factor receptors (VEGFRs), fibroblast growth factor receptor 4 (FGFR-4), and tyrosine kinase receptor for the hepatocyte growth factor (encoded by the *MET* proto-oncogene) often has been reported in MTCs [12]. The increased understanding in the molecular pathogenesis of MTC has led to the testing of several tyrosine kinase inhibitors (TKIs) specific for RET and other potential targets involved in angiogenesis [13,14]. Indeed, MTC is highly vascularized, and the impairment of tumor-induced angiogenesis represents an effective therapeutic approach [15,16,17].

Vandetanib (VAN) and Cabozantinib (CAB) are TKIs currently used for treating unresectable, progressive, and symptomatic MTCs. These drugs increase progression-free survival [18,19,20]. VAN (ZD6474) targets RET, VEGFR-2 and -3, FGFR, and EGFR [14,21,22]. CAB (XL184) is a small molecule targeting RET, VEGFR-2 and MET [23,24]. A head-to-head comparison in the same clinical trial between VAN and CAB has not been published yet. Two different phase III trials, ZETA and EXAM, evaluated the anti-tumor activity of VAN and CAB, respectively, versus a placebo [25,26]. However, several differences in study-design and enrolled populations make it difficult to make a direct comparison between these two drugs [27,28,29]. More recently, a retrospective multicenter study collected clinical data from a cohort of 48 patients with metastatic or locally advanced MTC who received treatment with VAN and/or CAB. Median progression-free survival for VAN and CAB were 17 and four months, respectively. However, the poorer prognosis of patients treated with CAB might be due to the high number of patients receiving CAB as second-line treatment after VAN treatment failure, when the course of the disease was more aggressive [30].

Lacking clinical trials directly comparing VAN versus CAB in patients with MTC, there are only a few in vitro studies evaluating the cell survival and anti-proliferative effects of these compounds in MTC cell lines [31,32]. Presently, comparative studies in vitro and/or in vivo on the effects of VAN and CAB on tumor-induced angiogenesis in MTC have not been published.

The zebrafish (*Danio rerio*) is a powerful animal model that has become an important preclinical tool particularly suitable for analyzing different aspects of tumor growth and progression, such as cell–stromal interactions, tumor-induced angiogenesis, and metastasis formation by performing xenotransplantation of human or mouse cancer cells in several sites of embryos [33]. Taking this context, we have recently developed an in vivo platform, based on xenotransplantation of neuroendocrine tumor cells in *Tg(fli1a:EGFP)^y^* transgenic fluorescent zebrafish embryos expressing EGFP (Enhanced Green Fluorescent Protein) under the control of the endothelial-specific gene promoter *fli1a* [34,35,36]. The implantation of MTC cell lines allows us to follow in vivo tumor-induced angiogenesis. Moreover, taking advantage of the permeability of zebrafish embryos to small molecules dissolved in their culture media, it is easy to study the anti-angiogenic activity of TKIs.

The aim of the present study is to compare in vitro and in vivo the anti-tumor activity of VAN and CAB in MTC, with a particular focus on angiogenesis through this innovative zebrafish model.

## 2. Results

### 2.1. Effects of Vandetanib (VAN) and Cabozantinib (CAB) on Cell Viability in Human Medullary Thyroid Carcinoma (MTC) Cell Lines 

Following six days of incubation, both vandetanib (VAN) and cabozantinib (CAB) significantly decreased the viability of TT (Figure 1a) (VAN, IC_50_: 1.5 × 10^−7^ M, maximal inhibition: −92.7%; CAB, IC_50_: 1.7 × 10^−7^ M, maximal inhibition: −91.2%) and MZ-CRC-1 (Figure 1b) (VAN, IC_50_: 1 × 10^−7^ M, maximal inhibition: −83.7%; CAB, IC_50_: 1.5 × 10^−7^ M, maximal inhibition: −74.9%) cell lines. Considering these results for further in vitro experiments, we selected the IC_50_ concentrations of VAN and CAB in MTC cells after six days of incubation. 

### 2.2. Effects of Vandetanib (VAN) and Cabozantinib (CAB) on Cell Cycle

Both drugs decreased significantly the fraction of TT cells in the S and G_2_/M phases after six days of incubation, with a more prominent effect after cabozantinib (CAB) (−59.5% and −22.3% versus the untreated control, *p* < 0.001 and *p* < 0.01, respectively) compared to vandetanib (VAN) (−31.4% and −12.5% versus the untreated control, *p* < 0.001 and *p* < 0.05, respectively) (Figure 2c,d); together with a concomitant accumulation of cells in G_0_/G_1_ (VAN: +29.3%, CAB: +20.1%, *p* < 0.05) (Figure 2b) and sub-G1 phase (VAN: +141%, CAB: +176%, *p* < 0.001) (Figure 2a). Regarding MZ-CRC-1 cells, VAN exerted a more potent effect in decreasing cells in S (−61.7%, versus the untreated control *p* < 0.001) and G_2_/M (−26.5%, versus the untreated control *p* < 0.05) phases (Figure 2g,h) compared to CAB, which significantly decreased the number of cells in the S phase (−34.2% versus the untreated control, *p* < 0.01) (Figure 2g). Similarly, both drugs increased the proportion of MZ-CRC-1 cells in the G_0_/G_1_ phase (VAN: +11.8%, versus the untreated control, *p* < 0.01; CAB: +9.6% versus the untreated control, *p* < 0.01) and the sub-G1 phase (VAN: +98%, *p* < 0.05; CAB: +91%, *p* < 0.01) (Figure 2e,f). 

### 2.3. Effects of Vandetanib (VAN) and Cabozantinib (CAB) on Apoptosis

Following six days of incubation, both drugs significantly increased the fraction of medullary thyroid carcinoma (MTC) cells in apoptosis. Cabozantinib (CAB) showed a more potent increase in the number of TT cells in early apoptosis (+1750% versus the untreated control, *p <* 0.001) and late apoptosis (+316% versus the untreated control, *p* < 0.001) compared to vandetanib (VAN) (+805% and +215% versus the untreated control, *p* < 0.001 and *p* < 0.01, respectively) (Figure 3a,b). There was no statistically significant change in the number of necrotic treated TT cells compared to the untreated control (Figure 3c). Regarding MZ-CRC-1, VAN markedly increased the fraction of cells in early apoptosis (+488%, versus the untreated control, *p* < 0.05), late apoptosis (+106% versus the untreated control, *p* < 0.05) and necrosis (+117% versus the untreated control, *p* < 0.01). A lower pro-apoptotic activity was observed with CAB, which increased the fraction of MZ-CRC-1 cells in early apoptosis (+396% versus the untreated control, *p* < 0.05), late apoptosis (+77% versus the untreated control, *p* < 0.05) and necrosis (+85% versus the untreated control, *p* < 0.01, respectively) (Figure 3d–f).

### 2.4. Effects of Vandetanib (VAN) and Cabozantinib (CAB) on Physiological Angiogenesis in Zebrafish Embryos

To test the anti-angiogenic potential of vandetanib (VAN) and cabozantinib (CAB) on the physiological angiogenesis of transgenic fluorescent zebrafish *Tg(fli1a:EGFP)^y1^,* we treated embryos with different concentrations of these tyrosine kinase inhibitors (TKIs) dissolved in the culture media. Following 24 h of treatment, we analyzed the development of the SIV (sub-intestinal vein) plexus, in particular we counted the number of vertical vessels comprising the SIV basket. Occurring at 72 h post fertilization (hpf) in the control embryos treated with vehicle dimethyl sulfoxide (DMSO), SIV developed as a basket-like structure over the yolk, blood vessels were lined in an orderly vertical pattern, and the integrity appeared to be well maintained (Figure 4a). Embryos treated with increasing concentrations of VAN and CAB showed a dose-dependent reduction in the number of vertical vessels and their size appeared narrower than those observed in the controls (Figure 4b–e). CAB displayed a more potent anti-angiogenic effect than VAN (Figure 4b–e). Occurring at 5 × 10^−6^ M, the highest tested dose for CAB, we observed a complete inhibition of the SIV vessel formation in embryos (Figure 4c,e) while, at this concentration, VAN (Figure 4b,d) only moderately reduced the number of vertical vessels in embryos compared to the control (Figure 4a).

### 2.5. Effects of Vandetanib (VAN) and Cabozantinib (CAB) on Tumor-Induced Angiogenesis in Zebrafish Embryos

We analyzed the anti-angiogenic potential of vandetanib (VAN) and cabozantinib (CAB) on TT and MZ-CRC-1 cell lines implanted in 48 hpf *Tg(fli1a:EGFP)^y1^* zebrafish embryos, taking advantage of the zebrafish medullary thyroid carcinoma (MTC) xenograft platform. Red dye-loaded MTC cells were grafted into the subperidermal space (between the periderm and the yolk syncytial layer) close to the sub-intestinal vein (SIV) plexus of 48 hpf embryos. Following only 24 h, grafted embryos showed vessel structures that sprouted from the SIV plexus toward the tumor mass (Figure 5).

Following the tumor cell implantation, injected embryos were treated with different concentrations of VAN and CAB. Following 24 h of treatment, the quantification of the tumor-induced vessel length revealed a reduction in vascular sprouts starting from the SIV plexus in a dose-dependent manner for both drugs, compared to dimethyl sulfoxide (DMSO)-treated embryos (Figure 6 and Figure 7). Particularly, CAB resulted in being more potent in inhibiting tumor-induced angiogenesis. Regarding both cell lines, the IC_50_ of CAB (4.6 × 10^−7^ M and 5.7 × 10^−7^ M in embryos xenotransplanted with TT and MZ-CRC-1, respectively) was significantly lower (*p* < 0.0001) than the VAN IC_50_ (2.5 × 10^−6^ M and 2.8 × 10^−6^ M in embryos xenotransplanted with TT and MZ-CRC-1, respectively) after 24 h of treatment. Moreover, CAB exerted a maximal anti-angiogenic effect (E_max_) that was significantly higher than VAN (*p* < 0.005) in zebrafish embryos injected with both cell lines (Figure 6g,h and Figure 7g,h).

## 3. Discussion

During the present preclinical study, we provide new findings about the anti-angiogenic effects of vandetanib (VAN) and cabozantinib (CAB), two tyrosine kinase inhibitors (TKIs) approved for the treatment of unresectable, progressive, and symptomatic medullary thyroid carcinomas (MTCs).

First, we perform an in vitro head-to-head comparison of these two drugs in the same experimental conditions using two human MTC cell lines to evaluate their anti-tumor activity and related mechanisms. Considering the literature, the anti-proliferative effects of VAN and CAB have been separately analyzed in several in vitro studies [37,38,39,40,41,42,43,44]. However, only a few in vitro studies have compared in the same experimental conditions the anti-tumor activity of VAN and CAB in MTC cell lines. Verbeek and colleagues reported a certain specificity of VAN and CAB for different *REarranged during Transfection* (*RET)* mutations in vitro, that differentially characterized human MTC cell lines, TT, and MZ-CRC-1. VAN inhibited cell proliferation at the lowest concentration in MZ-CRC-1 (IC_50_: 2.6 × 10^−7^ M), while CAB exerted the most effective inhibition in TT cells (IC_50_: 4 × 10^−8^ M) [31]. Regarding another in vitro head-to-head study, MTC cells were exposed to increasing doses of these two drugs for 48 h and then recovered in a drug-free fresh culture medium for 48 h before measuring their viability using an MTT assay. During these experimental conditions, VAN and CAB exerted a similar cell growth inhibition in both TT and MZ-CRC-1 cells [32]. Following six days of incubation with VAN or CAB, we found comparable inhibition of cell viability in both MTC cell lines. It is well known that these two TKIs are multitarget agents, therefore the anti-proliferative effects that we reported in our long-term treatment may be mediated by the simultaneous inhibition of multiple targets, in addition to REarranged during Transfection (RET) inhibition. 

These anti-proliferative activities were modulated by the cell cycle arrest and/or induction of apoptosis. It already has been demonstrated that VAN is mainly cytostatic in MTC cells [42,45] with a lock in the G_0_/G_1_ phase and no increase in apoptosis [41]. Conversely, CAB exerted a different modulation of cell cycle phases and had a pro-apoptotic effect in MTC cells [44,46]. During the head-to-head study by Starenki et al., after four days of incubation with VAN and CAB, both drugs (10^−6^ M) increased the sub-G1 phase in TT cells, whereas in the MZ-CRC-1 cell line they (5 × 10^−7^ M) decreased the S phase with a concomitant increase in the G_0_/G_1_ phase [32]. Regarding both MTC cell lines, we found that VAN and CAB induced a significant decrease in cells in the S and G_2_/M phases and an accumulation of cells in the G_0_/G_1_ phase after six days of incubation. Moreover, an increased sub-G1 phase after exposure with both drugs was predictive of a potential pro-apoptotic activity, which was confirmed through FACS analysis with Annexin/Propidium iodide (PI) staining of MTC cell lines.

Angiogenesis inhibition is another relevant anti-tumor mechanism of VAN and CAB. Indeed, both drugs inhibit the activity of various tyrosine kinases, which are implicated in angiogenesis. Currently, direct comparative studies in vitro and/or in vivo on the effects of VAN and CAB on tumor-induced angiogenesis in MTC have not been published.

Concerning different cell lines (such as PC3, MDA-MB-231, BaF3 and HUVEC), it has been demonstrated that CAB is a potent inhibitor of several tyrosine kinases involved in angiogenesis (MET, VEGFR-2, RET, KIT, AXL, TIE-2 and FLT-3). CAB exerted an anti-angiogenic effect in vitro, as shown by the inhibition of endothelial cell tube formation in HMVEC. To analyze in vivo the effect of CAB on tumor-induced vasculature, anti-angiogenic-sensitive human breast cancer MDA-MB-231 cells expressing *MET* and *VEGF* were implanted in mice. The drug administration increased hypoxia and cell death in tumor cells as well as in the endothelial cells of tumor-induced vasculature [23]. A recent work evaluated the anti-angiogenic effect of VAN and CAB on the stability of the vascular network using a microfabricated platform consisting of a micro-physiological system that incorporates human tumor cells in a 3D extracellular matrix, supported by perfused human microvessels, to create vascularized microtumors. Through this platform, the anti-angiogenic effects of several TKIs have been tested. Interestingly, CAB had more effective results than VAN, probably due to its activity against TIE-2, as well as to VEGFR-2 [47]. 

However, angiogenic assays consisting of in vitro cell-based models, in contrast to in vivo models, cannot simulate the biological complexity associated with blood vessels growing in their natural environment. Considering this context, zebrafish represents an ideal platform to analyze in vivo the angiogenesis process in physiological and pathological conditions. Several anti-angiogenic agents have been successfully tested in zebrafish [48]. Zebrafish embryos are permeable to small molecules, which can be dissolved in their culture medium; moreover, it is well known that some aspects of the physiological development of the sub-intestinal vein (SIV) and intersomitic and retinal vessels can simulate tumor-induced angiogenesis and be an important platform to test and quantify the effect of anti-angiogenic compounds [49]. Moreover, a zebrafish tumor-xenograft is a suitable platform to study neovascularization occurring with cancer progression in live animals. The implanted cells are able to form masses and recruit zebrafish endothelial cells that can infiltrate the tumor mass and lead to the formation of new vessels which express known endothelial markers, such as *VE-cadherin*, *fli1,* and *vegfr2* [50,51,52]. Noteworthy, the rapidity of this procedure and the response to angiogenesis inhibitors (24–48 h post treatment) makes this model a promising platform to perform preclinical drug screening [53]. 

Two different studies evaluated the anti-angiogenic effects of VAN and CAB in zebrafish separately [54,55]. Beedie and colleagues performed a dose response study with the aim of identifying potential risks of fetal toxicity in drugs that target the developing blood vessels of zebrafish and chicken embryos. Different TKIs were tested and VAN was identified among the less potent compounds, but it was still able to induce defects in vivo [54]. Recently, Wu and colleagues compared the anti-angiogenic and the anti-cancer potential of CAB and other drugs in a gastric cancer xenograft zebrafish model. Xenografted embryos treated with 5 × 10^−7^ M CAB showed a reduction of 15% in tumor-induced angiogenesis compared to the controls [55]. During a multiple screening, testing the effects of different anti-angiogenic compounds, zebrafish embryos were treated with different doses of VAN and CAB. The ability of anti-angiogenic agents to inhibit SIV physiological development after a 72 h treatment was evaluated, determining the anti-angiogenic efficacy for each compound. A more prominent effect of CAB at lower doses with respect to VAN (CAB: EC_50_ = 10^−7^ M and VAN: EC_50_ = 10^−5^ M) has been observed in this study [56].

Here, we compared in vivo the effect of VAN and CAB on physiological angiogenesis as well as on MTC tumor-induced angiogenesis, a first to our knowledge, taking advantage of a zebrafish xenograft model after the implantation of TT and MZ-CRC-1 cell lines. Analyzing their effects on the physiological angiogenic development, we found that both drugs inhibited SIV development in a dose-dependent manner. Following 24 h of incubation, CAB completely inhibited SIV development at lower concentrations compared to VAN. Regarding tumor-induced angiogenesis, *Tg(fli1a:EGFP)^y1^* zebrafish embryos were used as recipients for the xenotransplantation of MTC cell lines. Only 24 h after the implantation, grafted cells affected the physiological angiogenesis of the SIV plexus, leading to the formation of endothelial sprouts starting from the SIV toward the tumor mass. In our MTC xenograft model, both drugs showed tumor-induced angiogenesis inhibition in a dose-dependent manner. CAB had a stronger inhibitory effect on angiogenesis than VAN in embryos injected with both MTC cell lines.

In conclusion, through an innovative zebrafish model we found that the anti-angiogenic activity of CAB resulted in being more potent than that of VAN in MTC. Zebrafish have proven to be a powerful, reliable, and effective platform for the testing of anti-angiogenic compounds. Later, this zebrafish MTC xenograft model coupled with drug screening, may be a useful pre-clinical tool to enhance the understanding of molecular interactions between anti-angiogenic agents and different biological pathways involved in MTC progression.

## 4. Materials and Methods

### 4.1. Reagents and Cell Culture

Vandetanib (VAN), and cabozantinib (CAB) were provided by Cayman Chemicals (Ann Arbor, MI, USA). Stock solutions (4 mM) were made in 100% dimethyl sulfoxide (DMSO) and diluted with culture media before use. Two human medullary thyroid carcinoma (MTC) cell lines, TT and MZ-CRC-1, were kindly provided by Prof. Lips (Utrecht, the Netherland). Cells were maintained at 37 °C in 5% CO_2_ and cultured in T75 flasks filled with 10 mL of F-12K Kaighn’s modification medium (Gibco™ Thermo Fisher Scientific, Waltham, MA, USA). Media was supplemented with 10% heat-activated fetal bovine serum (FBS) (Invitrogen™ Thermo Fisher Scientific, Waltham, MA, USA) and 10^5^ U·L^−1^ penicillin/streptomycin (EuroClone™, Milan, Italy). Cells were harvested by trypsinization (Trypsin 0.05% and EDTA 0.02%) (Sigma-Aldrich^®^ Merck KGaA, Darmstadt, Germany), resuspended in complete medium, then counted through an optical microscope using a standard haemocytometer before plating. Cells used in all experiments were below 5 passages. All in vitro experiments were monitored for up to 6 days of drug incubation. We performed long-term treatments due to the slow doubling time (about 4 days) of the MTC cell lines.

### 4.2. Assessment of Cell Viability

Medullary thyroid carcinoma (MTC) cell lines were seeded in 96 well plates at their optimal culture concentration (TT: 20 × 10^3^ cells/well; MZ-CRC-1: 20 × 10^3^ cells/well). The following day, the cell culture medium was replaced with a medium containing vandetanib (VAN) and cabozantinib (CAB) for 3 days at different concentrations (from 5 × 10^−9^ to 10^−5^ M). Then, the medium was replaced with a new one containing drugs at the same differing concentrations for a further 3 days. A culture media containing an equivalent dimethyl sulfoxide (DMSO) concentration of the highest treatment dose served as the vehicle control. Following six days, an MTT assay (3-(4,5-dimethylthiazol-2-yl)-2,5-diphenyltetrazolium bromide) was performed, as previously described [57]. Considering these results, the IC_50_ were statistically calculated for each cell line using the Prism 5.0-GraphPad (GraphPad Software Inc., La Jolla, CA, USA).

### 4.3. Cell Cycle and Apoptosis Evaluation

Cell lines were seeded in 6-well plates in duplicates (MZ-CRC-1 and TT 3 × 10^5^ cells/well). The following day, the cell culture medium was replaced with a medium containing an equivalent dimethyl sulfoxide (DMSO) concentration (control), vandetanib (VAN), or cabozantinib (CAB) at their EC_50_ for 3 days. Then, the medium was replaced with a new one containing the vehicle or drugs with the same concentrations for a further 3 days, at the end of which the cells were harvested by gentle trypsinization, washed with cold PBS (calcium- and magnesium-free), and collected by centrifugation at 1200× *g* for 5 min. A propidium iodide (PI) (Sigma-Aldrich^®^ Merck KGaA, Darmstadt, Germany) solution (50 μg/mL PI, 0.05% Triton X-100 and 0.6 μg/mL RNase A in 0.1% sodium citrate) was added to stain the pellets at 4 °C for 30 min. To evaluate the cell-cycle PI for each tube, 10,000 cells were immediately measured and fluorescence was collected as FL1-A with a FACScalibur flow cytometer (Becton Dickinson, Erembodegem, Belgium) using Cell Quest Pro Software and data analyzed, as previously reported [57]. Cell cycle distribution was expressed as the percentage of cells in the G_0_/G_1_, S, and G_2_/M phases compared to the control. Regarding apoptosis, cells were resuspended in 100 μL of 1X binding buffer (BB: 1.4M NaCl, 0.1M HEPES/NaOH, pH 7.4, 25 mM CaCl2). Following incubation with 5 μL of Annexin V-FITC (BD Pharmingen, San Diego, CA, USA) and 10 μL PI (50 μg/mL in PBS) for 15 min at room temperature in the dark for each sample, 400 μL of 1X BB was added and stained cells were analyzed using FACScalibur on 10,000 events and analyzed, as previously described [57]. 

### 4.4. Zebrafish Line and Maintenance

Embryo and adult zebrafish (*Danio rerio*) were raised and maintained according to Italian (D.Lgs 26/2014) and European laws (2010/63/EU and 86/609/EEC). Embryos were staged according to morphological criteria [58]. Starting from 24 hpf, embryos were cultured in fish water (0.1 g/L NaHCO3, 0.1 g/L Instant Ocean, 0.192 g/L CaSO_4_•2H_2_O) containing 0.003% PTU (1-phenyl-2-thiourea; Sigma-Aldrich^®^ Merck KGaA, Darmstadt, Germany) to prevent pigmentation, and 0.01% methylene blue to prevent fungal growth. All experiments were performed on *Tg(fli1a:EGFP)^y1^* transgenic fluorescent zebrafish embryos [59]. 

### 4.5. In Vivo Subintestinal Angiogenesis Assay on Zebrafish Embryos

Occurring at 48 hpf, transgenic embryos were treated for 24 h with different concentrations of vandetanib (VAN) and cabozantinib (CAB). Stock solutions of VAN and CAB, prepared in dimethyl sulfoxide (DMSO), were diluted in fish water to concentrations ranging from 5 × 10^−7^ up to 1 × 10^−5^ M, and from 5 × 10^−8^ up to 5 × 10^−6^ M for VAN and CAB, respectively. As a control, some embryos were incubated with a fish medium containing the same concentration of DMSO used for the drug treated embryos. Occurring at 72 hpf, sub-intestinal vein (SIV) plexus images were taken with a Leica M205 FA stereomicroscope equipped with a Leica DFC 450 C digital camera using the LAS software (Leica Microsystems, Wetzlar, Germany). Vertical vessels that composed the SIV basket were counted in the controls and in the treated embryos to analyze physiological angiogenesis. 

### 4.6. In Vivo Zebrafish Assay for Tumor-Induced Angiogenesis

Forty-eight hours post-fertilization, zebrafish *Tg(fli1a:EGFP)^y1^* embryos were anesthetized with 0.016% tricaine (Ethyl 3-aminobenzoate methanesulfonate salt, Sigma-Aldrich^®^ Merck KGaA, Darmstadt, Germany) and implanted with TT and MZ-CRC-1 cells, as previously described [34,60,61]. Briefly, TT and MZ-CRC-1 cells were labeled with a red fluorescent viable dye (CellTrackerTM CM-DiI, Invitrogen™ Thermo Fisher Scientific, Waltham, MA, USA), resuspended with PBS, and grafted into the sub-peridermal space of *Tg(fli1a: EGFP)^y1^* embryos, close to the sub-intestinal vein (SIV) plexus. As a control of the implantation, we considered embryos injected with only PBS. This transplantable platform was used to test CAB and VAN effects on tumor-induced angiogenesis. Following the implantation, zebrafish embryos were treated for 24 h with these two drugs and directly dissolved into fish water. The drug concentrations ranged from 5 × 10^−7^ up to 1 × 10^−5^ M, and from 5 × 10^−8^ up to 2.5 × 10^−6^ M for VAN and CAB, respectively. The untreated controls were considered to be the injected embryos incubated in the fish water and the vehicle in which the experimental substance was dissolved (dimethyl sulfoxide (DMSO)). Assays were performed 3 times, considering about 20 embryos in each experimental group. All images were taken at 24 h post-injection with a Leica M205 FA stereomicroscope equipped with a Leica DFC 450 C digital camera using the LAS software (Leica Microsystems, Wetzlar, Germany). To measure the arbitrary unit of tumor-induced angiogenesis, we calculated the total cumulative length of the vessels sprouting from the SIV plexus and the common cardinal vein in each embryo using Fiji software.

### 4.7. Statistical Analysis

All experiments were performed at least three times. Statistical differences among groups were first evaluated using a *t*-test or ANOVA test together with the standard post hoc test (Newman–Keuls). A *p* value < 0.05 was considered significant. Statistical comparison of the logIC_50_ and maximal anti-angiogenic effect (E_max_) values were performed with the extra sum-of-squares F test approach (cutoff at *p* = 0.05). The values reported in the figures represent the mean ± standard error of the mean (S.E.M). Regarding statistical analysis, graphpad prism 5.0 was used (GraphPad Software Inc., La Jolla, CA, USA).

## Figures and Tables

**Figure 1 ijms-22-03031-f001:**
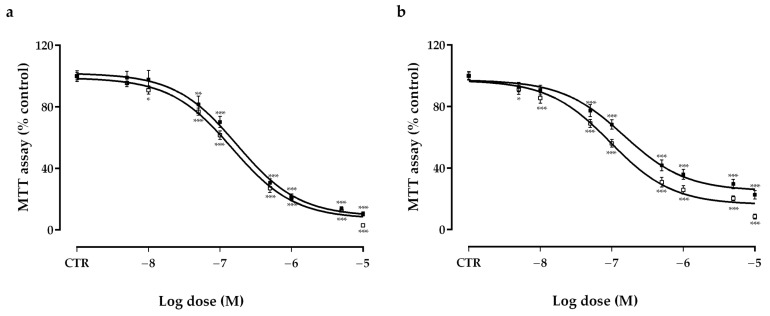
Effects of vandetanib (VAN) (□) and cabozantinib (CAB) (■) on cell viability in TT (**a**) and MZ-CRC-1 (**b**) cell lines measured using an MTT assay. Cells were incubated for six days with vehicle (control), or with the drug at different concentrations, as described in the Material and Methods. Dose response curves were expressed as a nonlinear regression (curve fit) of log (concentration drug) versus the percentage of the control. Values represent the mean ± S.E.M. of at least three independent experiments in six replicates. Control (CTR) values have been set to 100%. * *p* < 0.05; ** *p* < 0.01; *** *p* < 0.001.

**Figure 2 ijms-22-03031-f002:**
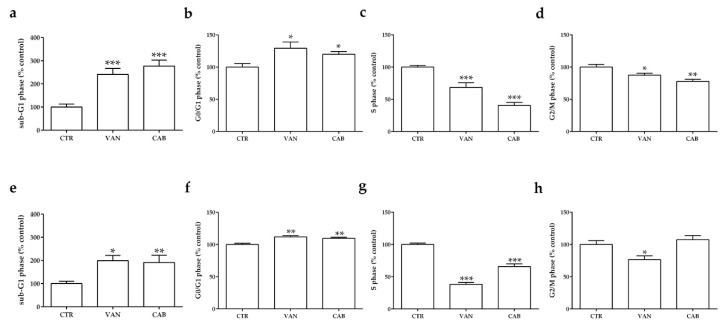
Effects of vandetanib (VAN) and cabozantinib (CAB) on the cell cycle in medullary thyroid carcinoma (MTC) cell lines. Cell cycle analysis after six days of incubation with VAN and CAB in TT (**a**–**d**) and MZ-CRC-1 cell lines (**e**–**h**). Cells were detected using FACS analysis after staining with propidium iodide. Control (CTR) values have been set to 100%. Cell cycle distribution is expressed as the percentage of cells in the G_0_/G_1_, S, and G_2_/M phases compared to the untreated CTR. Values represent the mean ± S.E.M. of at least three independent experiments. * *p* < 0.05; ** *p* < 0.01; *** *p* < 0.001.

**Figure 3 ijms-22-03031-f003:**
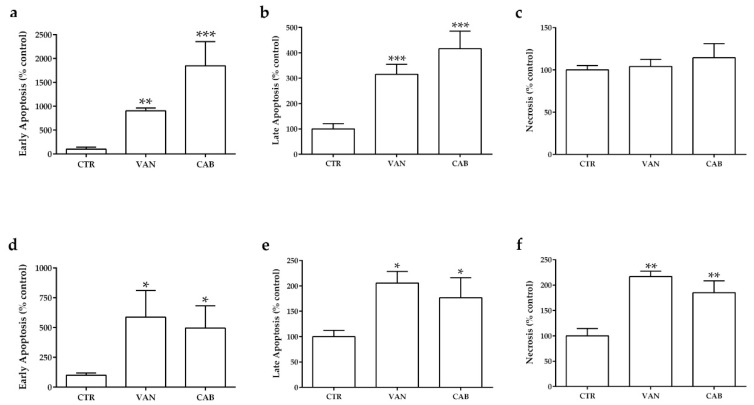
Effects of vandetanib (VAN) and cabozantinib (CAB) on apoptosis in medullary thyroid carcinoma (MTC) cell lines. Cell death analysis after six days of incubation with VAN and CAB in TT (**a**–**c**) and MZ-CRC-1 cell lines (**d**–**f**) through flow cytometry with Annexin V and propidium iodine. The proportions of cells in early apoptosis (**a**,**d**), late apoptosis (**b**,**e**) and necrosis (**c**,**f**) were expressed as the percentage compared with the untreated control (CTR) and represent the mean ± S.E.M. of at least three independent experiments. * *p* < 0.05; ** *p* < 0.01; *** *p* < 0.001.

**Figure 4 ijms-22-03031-f004:**
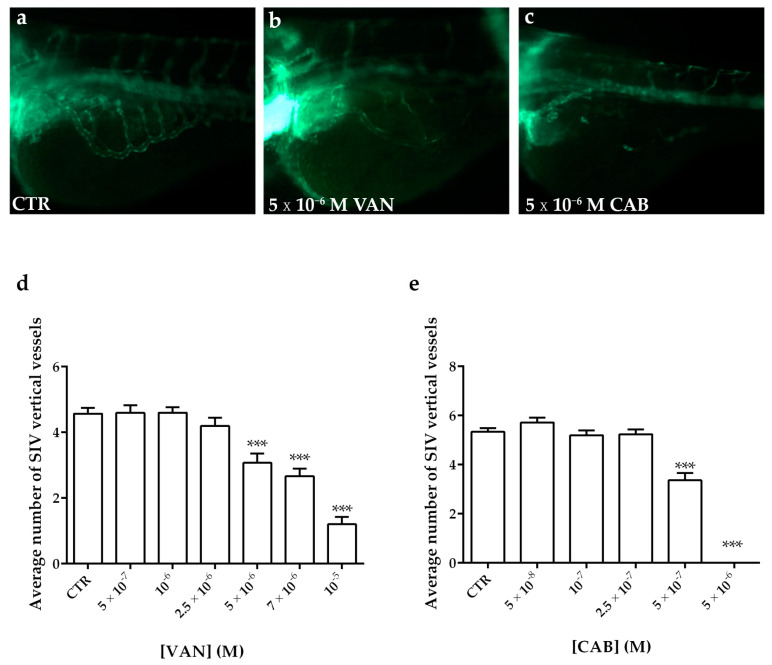
Anti-angiogenic effect of vandetanib (VAN) and cabozantinib (CAB) on the physiological development of the zebrafish sub-intestinal vein (SIV) plexus. Representative fluorescence images of the SIV basket of 72 hpf *Tg(fli1a:EGFP)^y1^* zebrafish embryos treated for 24 h with dimethyl sulfoxide (DMSO) (**a**), 5 × 10^−6^ M VAN (**b**) and CAB (**c**). Following 24 h of VAN (**d**) and CAB (**e**) treatments at different concentrations, the number of vertical vessels comprising the SIV basket was counted and compared with that of the control (CTR) embryos. Graphed values represent the mean ± S.E.M. *** *p* < 0.001 versus the CTR. Embryos are shown anterior to the left.

**Figure 5 ijms-22-03031-f005:**
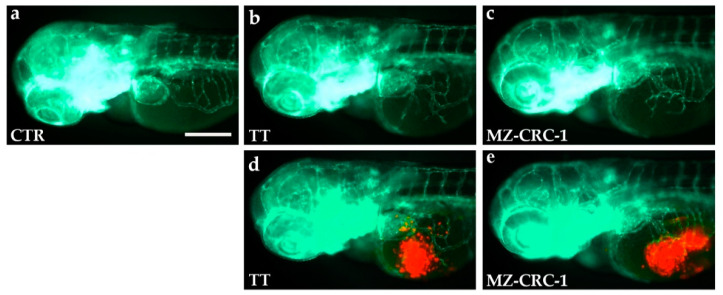
TT and MZ-CRC-1 implanted cells stimulate angiogenesis in zebrafish embryos after only 24 h post-injection. Representative fluorescence images of 72 hpf *Tg(fli1a:EGFP)^y1^* zebrafish control embryos (**a**) and embryos implanted at 48 hpf with TT (**b**,**d**) and MZ-CRC-1 cells (**c**,**e**). The red channel, corresponding to TT or MZ-CRC-1 cells, was omitted in panels b and c to highlight the tumor-induced microvascular network. Compared to the control, grafted embryos showed vessels in green that sprout from the sub-intestinal vein (SIV) toward the xenograft of both cell lines. Embryos are shown anterior to the left. Scale bar: 100 µm.

**Figure 6 ijms-22-03031-f006:**
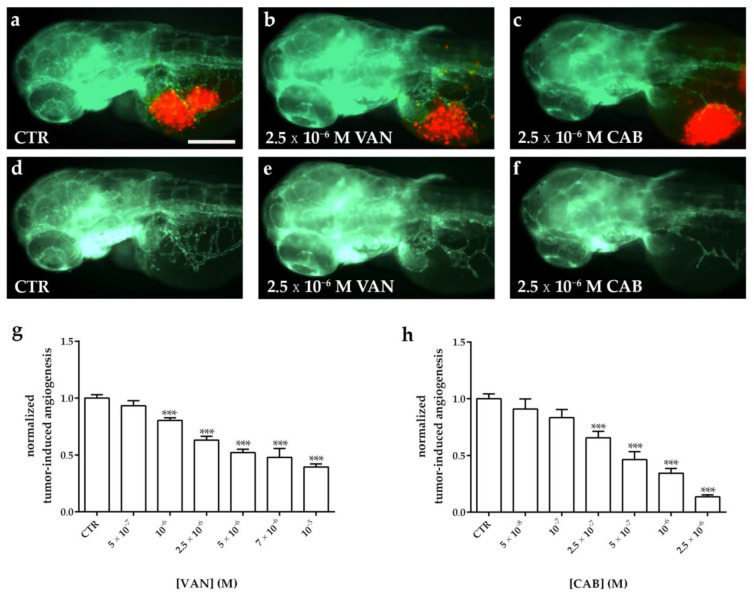
Effect of vandetanib (VAN) and cabozantinib (CAB) treatments on tumor-induced angiogenesis after TT cell xenograft in zebrafish embryos. Representative fluorescence images of 72 hpf *Tg(fli1a:EGFP)^y1^* zebrafish embryos implanted at 48 hpf with TT cells and subsequently treated for 24 h with dimethyl sulfoxide (DMSO) (**a**,**d**), VAN (**b**,**e**) and CAB (**c**,**f**). The red channel, corresponding to TT cells, was omitted in panels d, e, and f to highlight the tumor-induced microvascular network. Grafted larvae showed vessels in green that sprout from the sub-intestinal vein (SIV) toward the xenograft. Quantification of tumor-induced angiogenesis in TT-injected *Tg(fli1a:EGFP)^y1^* embryos after 24 h of VAN (**g**) and CAB (**h**) treatments at different concentrations. Control (CTR) values have been set to 1.0. Graphed values represent the mean ± S.E.M. *** *p* < 0.001 versus CTR. Embryos are shown anterior to the left. Scale bar: 100 µm.

**Figure 7 ijms-22-03031-f007:**
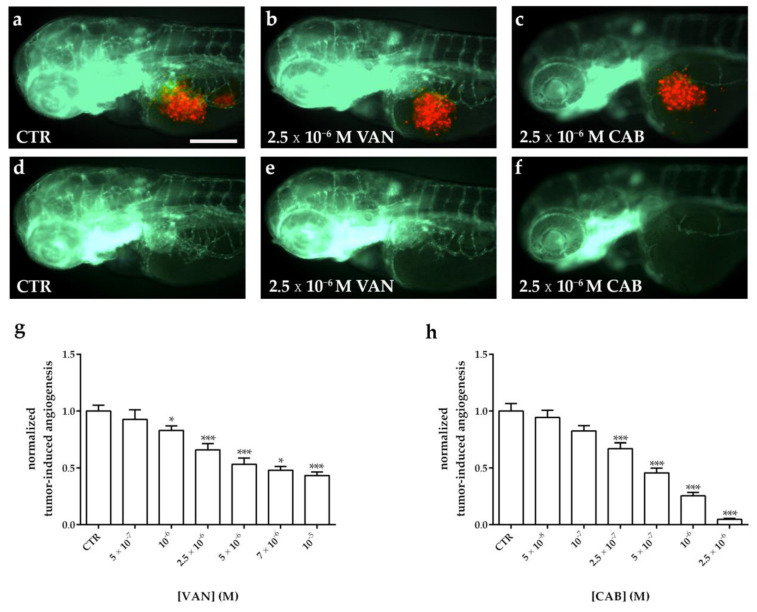
Effect of vandetanib (VAN) and cabozantinib (CAB) treatments on tumor-induced angiogenesis after MZ-CRC-1 cell xenograft in zebrafish embryos. Representative fluorescence images of 72 hpf *Tg(fli1a:EGFP)^y1^* zebrafish embryos implanted at 48 hpf with MZ-CRC-1 cells and subsequently treated for 24 h with dimethyl sulfoxide (DMSO) (**a**,**d**), VAN (**b**,**e**) and CAB (**c**,**f**). The red channel, corresponding to MZ-CRC-1 cells, was omitted in panels d, e, and f to highlight the tumor-induced microvascular network. Grafted larvae showed vessels in green that sprout from the sub-intestinal vein (SIV) toward the xenograft. Quantification of tumor-induced angiogenesis in MZ-CRC-1-injected *Tg(fli1a:EGFP)^y1^* embryos after 24 h of VAN (**g**) and CAB (**h**) treatments at different concentrations. Control (CTR) values have been set to 1.0. Graphed values represent the mean ± S.E.M. * *p* < 0.05 versus the CTR; *** *p* < 0.001 versus the CTR. Embryos are shown anterior to the left. Scale bar: 100 µm.

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
