# Peer review of "Vandetanib versus Cabozantinib in Medullary Thyroid Carcinoma: A Focus on Anti-Angiogenic Effects in Zebrafish Model"

_ijms, 2021, doi:10.3390/ijms22063031_

Round 1
Reviewer 1 Report
This is an interesting article comparing head to head the biological effect of vandetanib and cabozantinib on an innovative experimental model. Experimental data are scarce in the literature, thus the article presents interesting, original data, worthy of publication.
However, the following comments should be taken into account:
1) with regard to the adjective "antiangiogenic", the authors use several forms: “anti-angiogenic”, “antiangiogenenic”. It seems to me that the term "antiangiogenenic" is inappropriate and should be corrected especially in the description to Figure 5. Besides, I would propose to use a unified form of this word.
2) Figs. 2 and 3 - asterisks indicating statistical significance are not visible. Please correct this.
Clarification is also required for Figure 2 - not only in the text, but also in the description under the figure.
Author Response
1) with regard to the adjective "antiangiogenic", the authors use several forms: “anti-angiogenic”, “antiangiogenenic”. It seems to me that the term "antiangiogenenic" is inappropriate and should be corrected especially in the description to Figure 5. Besides, I would propose to use a unified form of this word.
Reply: We thank the reviewer for the comment. Following the reviewer’s suggestion we revised our manuscript accordingly, using “anti-angiogenic” term.
2) Figs. 2 and 3 - asterisks indicating statistical significance are not visible. Please correct this. Clarification is also required for Figure 2 - not only in the text, but also in the description under the figure.
Reply: We changed figure 2 and 3, correcting asterisk size and we added explicative information in the M&M section and in the figure 2 legend.
Reviewer 2 Report
The authors present an interesting study on the role of VAN and CAB that were found to have comparable effects on viability and cell cycle in medullary thyroid cancer cell lines carrying a RET mutation. They also used a zebrafish model to study the role of the 2 drugs in inhibiting the angiogenesis.
The study is correctly planned and the results are well described.. However some points should be discussed:
- The first question is related to the lenght of the drug incubation. Experiments have been run for 6 days. How did the authors decide use time time period?
- The effect of CAB and VAN on MTC cell lines should be mediated by the effect of these 2 drugs on RET. Very interestingly, no differences have been observed between the 2 cell lines that carry 2 different RET mutation. Can the authors make a hypothesis? Are the C634R and M918T RET mutations equally affected by the drugs?
Author Response
1) The first question is related to the lenght of the drug incubation. Experiments have been run for 6 days. How did the authors decide use time time period?
Reply: The choice of a long-term incubation (6 days) was related to the slow doubling time (about 4 days) of both MTC cell lines. A sentence has been added in M&M section.
2) The effect of CAB and VAN on MTC cell lines should be mediated by the effect of these 2 drugs on RET. Very interestingly, no differences have been observed between the 2 cell lines that carry 2 different RET mutation. Can the authors make a hypothesis? Are the C634R and M918T RET mutations equally affected by the drugs?
Reply: In our experimental conditions (drug exposure of 6 days with a renewal of drugs after 3 days), we did not observe any significant difference between CAB and VAN in TT (RET C634R mutation) and MZ-CRC-1 (RET M918T mutation) cell lines.
A similar antiproliferative effect has been previously reported by Starenki D et al. Cancer Biol Ther 2017. Taking into consideration that both compounds are multitarget tyrosine kinase inhibitors (VAN: RET, VEGFR-2 and -3, FGFR and EGFR; CAB: RET, VEGFR-2 and MET), it is difficult to conclude that the C634R and M918T RET mutations are equally affected by the drugs. Indeed, we cannot exclude the influence of other RET-independent pathways modulated by VAN and CAB. A short sentence has been included in the discussion section.